# Modulation of Oxidative Stress and Antioxidant Response by Different Polyphenol Supplements in Five-a-Side Football Players

**DOI:** 10.3390/nu15010177

**Published:** 2022-12-30

**Authors:** Lucrecia Carrera-Quintanar, Lorena Funes, María Herranz-López, Néstor Vicente-Salar, Raul Bonet-García, Carles Blasco-Peris, Vicente Micol, Antoni Pons, Enrique Roche

**Affiliations:** 1Food Science Laboratory, Department of Human Reproduction, Growth and Child Development, University of Guadalajara (CUCS), Guadalajara 44340, Mexico; 2Institute of Research, Development and Innovation in Healthcare Biotechnology of Elche (IDiBE), Miguel Hernández University (UMH), 03202 Elche, Spain; 3Department of Applied Biology—Nutrition, Institute of Bioengineering, Miguel Hernández University (UMH), 03202 Elche, Spain; 4Institute for Health and Biomedical Research (ISABIAL), 03010 Alicante, Spain; 5Department of Physical Education and Sport, University of Valencia, 46010 Valencia, Spain; 6CIBER Fisiopatología de la Obesidad y la Nutrición (CIBEROBN), Instituto de Salud Carlos III (ISCIII), 28029 Madrid, Spain; 7Research Group on Community Nutrition and Oxidative Stress, University of Balearic Islands, 07122 Palma, Spain

**Keywords:** antioxidant enzymes, futsal, inflammation, malondialdehyde, protein carbonyls

## Abstract

Oxidative stress is associated with playing soccer. The objective of the present report was to study the influence of different polyphenolic antioxidant-rich beverages in five-a-side/futsal players. The study was performed with a no supplemented control group (CG) and two supplemented groups with an almond-based beverage (AB) and the same beverage fortified with *Lippia citriodora* extract (AB + LE). At day 22, participants played a friendly futsal game. Blood extractions were performed at the beginning of intervention (day 1), before and after match (day 22) to determine oxidative stress markers and antioxidant enzyme activities in plasma, neutrophils and peripheral blood mononuclear cells (PBMCs). Malondialdehyde increased significantly in controls after the match in neutrophils, PBMCs and plasma compared to pre-match. Protein carbonyls also increased after the match in plasma in CG. In addition, malondialdehyde levels in neutrophils were significantly lower in the supplemented groups compared to controls. Post-match samples showed significant increases in neutrophil antioxidant activities in CG. Supplemented groups displayed variable results regarding neutrophil antioxidant activities, with superoxide dismutase activity significantly lower than in controls. Finally, post-match myeloperoxidase activity increased significantly in controls compared to pre-match and supplemented groups. In conclusion, polyphenolic antioxidant and anti-inflammatory supplements could be instrumental for optimal recovery after high intensity futsal games.

## 1. Introduction

Soccer is the most popular sports discipline worldwide. As such, soccer is played in many social segments and has led to the appearance of many game variants including soccer sevens, indoor football and beach soccer and futsal (five-a-side football). In all cases, basic soccer actions include running, sprinting, jumping, changes in direction and variations in rhythm [1]. To achieve this, players need to develop and train for strength, speed, agility, stability, ability and endurance, among others [2]. However, soccer practice is highly associated with muscle microinjury due to eccentric actions, accidental trauma, forced running and repetitive intermittent actions [3]. All these events cause inflammatory damage that is strongly related to oxidative stress [4,5]. Depending on the level of damage, the inflammation can lead to major muscle damage, limiting performance and requiring medical treatment [3]. In addition, high intensity physical exertion during the execution of specific intermittent actions results in oxygen overconsumption with the associated production of reactive oxygen species (ROS) [6]. High ROS production during exercise can cause oxidative damage to cellular macromolecules, such as lipids, proteins and nucleic acids [7,8,9]. Muscle damage and ROS production result in incomplete recovery, impaired performance and increased risk of lesions [10].

Therefore, the conditions observed during training sessions and matches in professional soccer practice strongly indicates that this is a sport discipline that can challenge antioxidant defense systems [11,12,13]. Subsequently, athletes in this sport discipline are candidates for antioxidant supplementation, although there is little information about this topic. In this vein, the practice of regular exercise favors the adaptation of endogenous antioxidant systems at the level of gene expression [14,15]. However, ROS overproduction after high intensity exercise, such as in the case of soccer, can overwhelm endogenous antioxidant defenses, compromising muscle repair and postexercise recovery [10]. In these cases, antioxidants provided in the form of supplements, such as vitamins C and E or polyphenols could increase the effect of intracellular antioxidants [16]. However, high doses of antioxidant supplements taken over long periods of time can impair the antioxidant adaptive response of specific circulating cell types by avoiding the optimal expression of genes coding for antioxidant enzymes [15,16]. Therefore, the antioxidant effect of supplements depends on the dose and the time of intake. In this respect, high doses of antioxidants are recommended when ROS are produced at high levels, to prevent extensive damage to the major intracellular macromolecules [17]. On the other hand, ROS production cannot be completely annulled by high doses of antioxidant consumption over long periods of time [18]. This is because ROS are instrumental agents in the transduction pathway that culminates in the activation of genes coding for antioxidant defenses, resulting in an optimal adaptation to exercise [5,19]. In this context, the Nrf2-Keap1 signaling plays a key role. Keap1 is an inhibitor of Nrf2 in basal conditions. However, in oxidative stress situations, Nrf2 is released from Keap1, translocated to the nucleus and heterodimerized with small Maf proteins. The dimers recognize enhancer sequences in the regulatory regions of Nrf2 target genes coding for antioxidant and metabolic enzymes [20].

One instrumental limitation of soccer is that it is a team sport, and each player has a different role in the field during the game, thereby exhibiting different anthropometric characteristics, performance and skills [21,22]. This implies that the energetic expenditure and the probability of undergoing muscle injuries and oxidative stress are dependent on the field position. In addition, the particular characteristics of each match can introduce different stress demands to players related to weather conditions, game strategy and psychological pressure, among others [23,24,25]. Altogether, this makes it difficult to homogenize experimental designs when performing intervention studies.

The objective of the present report was to study the influence of different beverages enriched in antioxidants on the modulation of oxidative stress in soccer players compared to nonsupplemented counterparts. Both beverages have been tested in previous reports [16,26]. To minimize the influence of field position and match variables, we performed the intervention in one friendly five-a-side football match, in which all field players played the same day and exerted similar efforts during the match.

## 2. Materials and Methods

### 2.1. Participants and Protocol

Thirty male federated players of the university five-a-side football league were recruited and distributed into 3 groups of 10 individuals each, according to anthropometric criteria from ISAK (International Society for Advancement of Kinanthropometry) recommendations [27]. Regarding body composition, no significant differences were found between participants in the different groups (Table 1). Within each group of 10, players were broken in 2 teams with 4 field players and a goalkeeper. The control group (CG) was the nonsupplemented group, consuming only water as beverage. The second group consumed an almond beverage (AB group) enriched with vitamins C and E for 22 days, which was used in previous studies [16,26,28]. The last group was supplemented during the same period of time (22 days) with the same beverage but enriched with *Lippia citriodora* extract (AB + LE group), which was used in previous studies [16,26]. As inclusion criteria, participants were active players who had played for at least one season in the university league, were free from chronic diseases, did not smoke, followed the nutritional plan prepared by the research team and assisted in the training sessions 5 days a week. Exclusion criteria were being unable to assist in the training sessions or follow the diet and consumption of sports supplements or prescribed drugs at the moment of intervention. Participants were informed about the objective of the intervention and signed a written consent form. The study was in accordance with the Helsinki Declaration for research on human beings, and was approved by the local Ethics Committee with reference IB 544/05 PI.

Over the course of 21 days, participants took part in 90-min training sessions for 5 days each week. Training sessions consisted of active warm up and dynamic stretching (15 min), different exercises of ball control, first touch, shooting and changes in rhythm and direction (40 min), a friendly match with game simulations (30 min) and relaxation (5 min). No muscle injuries occurred during the 21-day training period. Participant’s diet was designed using Dietsource software (Novartis, Barcelona, Spain) and adapted to interval routines: 60% carbohydrates, 25% lipids and 15% proteins. Daily energy intakes were estimated to be ~2100 kcal for resting days and ~2600 kcal for training days, taking into account the resting metabolism + thermal effect of food + physical activity expenditure. Resting metabolism was calculated according to Harris-Benedict equation. The thermal effect of food was estimated as 8.5% of the sum of resting metabolism plus exercise expenditure. Physical activity expenditure was estimated according to [29,30]. Diet adherence was supervised twice a week for each group of participants, following a serving method in a pocket document designed by our group.

Daily vitamin C intakes coming from diet were ~120 mg for resting days and ~325 mg for training days in all groups. Daily diet vitamin E intakes were ~5 mg for resting days and ~9 mg for training days in all groups. The beverage consumed during the whole intervention by the AB group consisted of a mixture of crushed almonds and orange juice supplemented with 50 mg of vitamin C/100 mL juice and 20 mg of vitamin E/100 mL juice. The beverage provided to the AB + LE group was the same as that for the AB group, but contained an additional 400 mg of *Lippia citriodora* extract (Monteloeder SL, Elche, Spain). Participants in both groups drank 500 mL/day of the corresponding beverages. The beverages were packed by Liquats Vegetals SL (Viladrau, Gerona, Spain) in white tetra bricks displaying only the expiration date. Therefore, the antioxidant potential of the beverages relies on the contents of vitamins C and E, polyphenols from crushed almonds/orange juice (AB group) and the presence of the *Lippia* extract (AB + LE group). Beverages were consumed at the end of the training session. The resting days beverages were consumed at home by each player.

On the last day of the study (day 22), the 2 teams of each group played a five-a-side football match lasting for 40 min real time (2 half-times of 20 min each), resting 5 min between each half-time. To avoid accidents, participants were not wearing accelerometers during the match. However, at minutes 7, 14 and 20, the match was stopped and the heart rate of participants was determined with Polar RS-800 accelerometers (Barcelona, Spain). Heart rate was not assessed in the goal keepers, due to the more static field position compared to the other players (Table 1).

### 2.2. Blood Sampling and Determination of Circulating Parameters

Blood samples were obtained from the antecubital vein after overnight fasting in EDTA vacutainers at the moment of recruitment (Day 1). Extraction on Day 1 was performed to verify that participants were in a healthy condition and to monitor tissue circulating markers before starting the training phase. On the day of the match (Day 22), participants had a lunch 2 h before the match rich in carbohydrates (8–10 g/kg body weight) in order to cover energy demands. Then, 10–12 mL of blood were extracted 30 min before and 30 min after the match under nonfasting conditions. Erythrocytes, neutrophils, peripheral blood mononuclear cells (PBMCs) and plasma were purified as indicated in [16].

Blood analysis was performed to check the health status of participants. Since the conditions for blood extractions were not similar (fasting vs. non fasting), we focused only in lactate and muscle damage parameters. Creatine kinase (CK), myoglobin, aspartate aminotransferase/serum glutamic oxaloacetic transaminase (AST/GOT) and alanine aminotransferase/serum glutamic pyruvic transaminase (ALT/GPT) were determined according standard laboratory procedures as indicated in [16].

### 2.3. Determination of Antioxidant and Myeloperoxidase Enzymatic Activities and Oxidative Stress Markers

Neutrophil and PBMC catalase (CAT), glutathione peroxidase (GPX), glutathione reductase (GRD) and superoxide dismutase (SOD) activities were measured on a microplate reader (SPECTROstar Omega, BMG LabTech GmbH, Offenburg, Germany) at 37 °C as previously indicated [31]. Myeloperoxidase (MPO) activity was determined by guaiacol oxidation as described in [32]. Protein carbonyls were determined in neutrophils, PBMCs and plasma, as described in [31]. Malondialdehyde (MDA) was determined by HPLC as described in [32].

### 2.4. Statistical Analysis

Statistical analysis was carried out using SPSS-26 software for Windows (IBM, Chicago, IL, USA). Data were tested for normality according to the Shapiro–Wilk test. Results were expressed as the mean ± SEM (standard error of the mean). Student’s *t*-test was used for intragroup analysis at different moments of intervention, as well as for comparison between groups. Values with a *p* < 0.05 were considered statistically significant.

## 3. Results

The aim of the present study was to analyze the antioxidant effect of different polyphenolic compounds in university-level five-a-side football players. We have analyzed the modulation of the activities of endogenous antioxidants, such as SOD, CAT, GPX and GRD in neutrophils and PBMCs after a friendly match. The results were compared to the baseline condition at the beginning of the study when students were returning after a time of inactivity during the summer season. The post-match results were also compared with the pre-match condition after 21 days following the same training sessions for all individuals. The antioxidants provided to volunteers have been developed in our laboratory and extensively characterized in previous studies [16,26,28,33,34,35].

Blood samples were obtained during the first day of the study under fasting conditions. The results indicated that all individual’s parameters were in the healthy range (data not shown). However, it cannot be compared these results to those obtained from blood analyses performed before and after the match, which were both performed under nonfasting conditions. To cover the high energy demands and avoid muscle lesions and fatigue, we allowed a rich carbohydrate breakfast 2 h before the match and isotonic beverages before, during and after the match. For these reasons, we only focused our analysis on lactate and muscle damage parameters (CK, myoglobin, AST/GOT and ALT/GPT) (Table 2). Compared to day 1, CK increased significantly before the match in all groups, likely as a result of the training sessions over 21 days (Table 2). Myoglobin displayed a tendency to increase in all groups (Table 2), but differences were not significant in pre-match situation. The muscle and hepatic markers AST/GOT and ALT/GPT were significantly increased only in the CG and AB + LE groups, respectively, before and after the match when compared to Day 1 (Table 2). After the match, lactate increased significantly in all groups, confirming the anaerobic component over the game (Table 2). Significant increases in CK and myoglobin are observed in CG, AB and AB + LE groups confirming muscle damage after the match respect pre-match values (Table 2).

The extent of oxidative damage in neutrophils, PBMCs and plasma was assessed by determining MDA and protein carbonyls as markers of oxidized lipids and proteins, respectively. The comparison of these parameters at the beginning of intervention vs. the pre-match condition could be indicative of the oxidative damage produced during the 21-day training period.

The CG presented a significant increase in plasma MDA and protein carbonyls at the end of the training period (Table 3). No changes were observed in neutrophils and PBMCs in the CG, suggesting an adequate intracellular antioxidant response (Table 3). Regarding the supplemented groups, we observed a significant decrease in plasma MDA and protein carbonyls in the AB group compared to the beginning of the intervention and with respect to the CG. On the other hand, the AB + LE group presented no changes in plasma MDA and protein carbonyls with respect to Day 1, although there was a significant reduction compared to the CG (Table 3). No significant oxidative changes were observed in neutrophils and PBMCs of the AB and AB + LE groups compared to the baseline condition (Day 1), except for a significant decrease in neutrophil MDA in the AB group compared to Day 1 and the CG (Table 3).

The determination of the oxidative markers after the match compared to the baseline condition could be indicative of the extent of oxidative damage caused during the game. Table 3 shows that MDA increased significantly after the match in the neutrophils, PBMCs and plasma of the CG. Plasmatic protein carbonyls were also significantly increased in the CG. However, in the AB group both oxidative markers were elevated only in plasma and remained unchanged compared to baseline conditions in neutrophils and PBMCs, except for MDA in neutrophils which was significantly lower than the CG. This significant change compared to the CG suggests a working intracellular antioxidant response in the AB group, with the plasma being less protected against oxidative stress in the CG and AB groups. Finally, the pattern observed in the AB + LE group was similar to the observed in the AB group, except for higher protein carbonyls in pre-match situation.

The next step was to decipher if the oxidative pattern reflected in Table 3 could be due to the activity of antioxidant enzymes or to the antioxidant effect of the beverages as observed previously in resistance training athletes [16]. No significant differences were observed in antioxidant enzyme activities at the beginning of the intervention (Day 1) in neutrophils and PBMCs in any groups (Table 4). MPO, a marker of inflammation [36], was not significantly different between groups (Table 4). Blood samples extracted before the match indicated no significant changes compared to the baseline condition in the activities of oxidative enzymes determined in neutrophils and PBMCs in the CG (Table 4). A similar pattern was observed in the AB and AB + LE groups, except for a significant increase in catalase in neutrophils (Table 4). Compared to the CG, SOD presented a decreased activity in the AB and AB + LE groups (Table 4). In addition, the neutrophil SOD activity in the AB + LE group was significantly lower compared to day 1 (Table 4). Furthermore, catalase activity increased significantly in neutrophils of AB and AB + LE groups compared to the CG (Table 4). No significant differences were observed in PBMCs before the match (Table 4), whereas MPO activity was significantly decreased in the AB and AB + LE groups compared to the baseline situation and to the CG before the match (Table 4).

Finally, blood samples obtained immediately after the match presented a significant increase in neutrophils for practically all antioxidant activities in the CG (Table 4). In the AB group, a significant increase was observed in CAT and GPX after the match compared to day 1 (Table 4). In the AB + LE group, post-match GPX activity was significantly increased when compared to baseline conditions. On the other hand, post-match SOD activity was significantly decreased compared to the baseline conditions in the AB + LE group (Table 4). In addition, SOD activity after the match was significantly decreased in the AB and AB + LE groups compared to the CG, and significantly lower in AB + LE compared to AB (Table 4). GPX activity was significantly increased in the AB + LE group in the pre-match compared to the AB, and in the post-match situation compared to CG (Table 4). No significant changes were observed in the enzymatic activities determined in PBMCs after the match, although they displayed a general tendency to increase (Table 4). Finally, post-match MPO activity increased significantly in the CG compared to Day 1, and was significantly decreased in the AB and AB + LE groups compared to the CG (Table 4).

## 4. Discussion

A first interpretation of the obtained results may suggest that the five-a-side football or futsal is a sports discipline prone to oxidative stress and inflammatory damage as observed previously in classical soccer [12]. In this context, the number of variables that could modulate oxidative stress responses in soccer seems to be wider than in other sport activities [37]. These include the environmental conditions (temperature, humidity) that change dramatically during the season, the variable demanding efforts when facing different opposing teams, the possibility to undergo muscle damage due to unexpected game actions and psychological motivation during the different matches, among others [1,2]. Overall, this implies that it is challenging to homogenize experimental conditions in this group of sports disciplines, making it very difficult to compare research studies. Despite this, in this particular intervention, we have tried to homogenize certain experimental conditions to introduce fewer variables. First, matches were played the same day with very similar stable and mild weather conditions. Second, and since the intervention was performed at the beginning of the season with no official matches played yet, all teams presented a similar performance level and psychological motivation status. This should imply similar demanding efforts of all teams during the match, according to no significant differences in heart rate between groups (Table 1). Nevertheless, not all variables that influence oxidative stress at different levels (running distance, changes in speed, accidental trauma, among others) were monitored. This could be considered a limitation of the study.

Nevertheless, this study presented other limitations to take into account when interpreting the results. First, the match played by participants was a friendly game because all participants belonged to the same university team. This could result in a lower intensity during the game and lower oxidative and inflammatory damage compared to an official competition match where demanding intensity must be higher [38]. However, the differences between opponent teams and game conditions during the season submit players to different stress conditions. In other words, players undergo different physiological and psychological tensions in each official match of the season, and comparison of the stress underwent in one match cannot be experienced to others. Therefore, we agree that the friendly match is played with a lower intensity than a competition match, but this intensity seemed at least to be quite homogeneous in all teams that participated in the study, according to the heart rate measured during the different matches.

A second limitation comes from the different roles of players during the game that could affect the oxidative stress and inflammation parameters [10]. Among the 5 participants on each team, the 4 field players performed with a similar effort, according to their heart rates, while the goal keeper´s heart rate was more stable. This could imply that the goal keepers would display less oxidative damage than field players due to their less intensive role during the game. However, the differences were not significant when compared to the rest of the players on the same team. Our first explanation is that in five-a-side football, goal keepers, although apparently static, have a more active participation than in other football variants. In addition, goal keepers experience more traumatic and contact actions contributing to increase inflammation [10]. This is reflected in slightly higher MPO values of the 2 goal keepers of CG compared to field players in the same group (Mean MPO activity = 88.7 µkat/10^9^ cells). However, this result must be interpreted with caution due to the low sample size. Studies on inflammatory damage focusing on this particular game position (goal keepers) and comparing other positions and soccer variants need to be performed in the future.

According to the oxidative stress markers, plasma seems to be the body compartment that presents more oxidative markers than neutrophils and PBMCs, as previously observed [12]. A possible interpretation is that plasma displays fewer antioxidant defenses than circulating cells. This explains why after the training period, and mainly after the match, MDA and protein carbonyls increased significantly in plasma in the CG. However, the supplemented groups presented lower values for plasma MDA and carbonyls than the CG, suggesting an antioxidant effect of the supplements on the plasma metabolites derived from the antioxidant molecules (vitamins or polyphenols). Regarding circulating cells, neutrophils and PBMCs presented significant MDA increases after the match only in the CG, while these cells seemed to be protected in the supplemented groups, again suggesting an antioxidant effect of the supplements. Neutrophils are implicated in the oxidative burst that appears when muscular damage occurs and are more exposed to oxidative damage [26]. The existence of muscle damage is verified by the increased post-match circulating levels of CK and myoglobin. Circulating levels of these markers were very similar suggesting a homogeneous intensity of players during the friendly match.

The response against the abovementioned oxidative markers may depend on the antioxidant response. This response may rely on the intracellular antioxidant activities of neutrophils in the CG. In fact, all antioxidant enzyme activities increased after the match in the CG to restore oxidative damage compared to both pre-match or baseline conditions. Similar results (increased SOD and CAT activities) were observed in a group of adolescent soccer players after six-month training program [13]. However, low levels of glutathione were observed in this study at the end of the 6-month training period, likely affecting the activities of glutathione-dependent enzymes. Nevertheless, we did not observe a decrease in the activities of GPX and GRD after 22 days of training. Our interpretation is that longer training periods (6 months vs. 20 days) could challenge the activities of glutathione antioxidant systems. Therefore, supplements that can enhance the activities of GPX and GRD could be instrumental for recovery during a soccer season [35]. Additional studies need to be performed to support this hypothesis.

On the other hand, the antioxidant response observed in neutrophils of the CG was not reproduced in the supplemented groups. Only neutrophil CAT and GPX activities increased in AB and AB + LE groups, but not SOD activity. One explanation for this observation is that these supplements reduce SOD gene expression (Cu-Zn-SOD and Mn-SOD), particularly AB + LE, as we have previously observed [16]. This reduction could condition the SOD enzymatic activity in the supplemented groups to present low values and a reduced removal rate of superoxide anion. Under these conditions, it has been described that overproduction of superoxide anion increases CAT and GPX activities in neutrophils [39]. However, it has been described that glutathione levels decrease after a soccer match [38]. Although we could not determine glutathione levels in the present intervention, we have demonstrated in previous reports that the polyphenolic compounds (verbascoside in particular) present in the *Lippia citriodora* extract can increase the activity of the glutathione dependent enzymes likely by preserving intracellular glutathione levels [35]. For this reason, *Lippia citriodora* extract was selected to complete the antioxidant action of the almond beverage. Moreover, and taking into account that the oxidation markers do not increase in the supplemented groups, we suggest that the intracellular antioxidant defenses could be reinforced by the supplement intake. In any case, further research is necessary to address this point and characterize the antioxidant network operating during the recovery period.

Finally, MPO activity was lower in the supplemented groups than in the CG in pre-match situation, and these low values were maintained after the match, suggesting a less inflammatory damage due to the game actions [26]. We propose that the candidates to modulate the inflammatory response could also be due to the polyphenols present in both drinks from crushed almonds/orange juice (AB and AB + LE) and Lippia extract (AB + LE). In addition to their antioxidant potential, it is well known that polyphenols are active anti-inflammatory agents [26,40,41]. This property of polyphenols is of particular interest in soccer, where trauma and violent contacts that can cause muscle damage, are common during the game (Souglis et al., 2018) [10]. Nevertheless, the contribution of other components present in the almond beverage (proteins and unsaturated fat) could be considered for future research.

## 5. Conclusions

Soccer and its variants are sports disciplines in which oxidative stress and inflammation appear very often. The main conclusions from the study are:-Increase of MDA and carbonyls in CG group after the match. MDA in neutrophils decreased in the supplemented groups.-Increased neutrophil antioxidant activities in the control group in the post-match situation.-SOD activity in supplemented groups was significantly lower compared to controls.-Increase in the post-match MPO activity compared to pre-match in CG that was attenuated in the supplemented groups.

Considering all these observations, antioxidant and anti-inflammatory supplements can be instrumental in improving post-match recovery, particularly after high intensity games. In addition, antioxidant supplements must reinforce and not interfere with the intracellular antioxidant response.

## Figures and Tables

**Table 1 nutrients-15-00177-t001:** Age, anthropometric parameters determined at the moment of recruitment (Day 1) and mean heart rate during the match (Day 22) determined in the field for players in control (CG), almond beverage (AB) and almond beverage + Lippia extract (AB + LE) groups.

Parameter (Units)	CG	AB	AB + LE
n	10	10	10
Age (years)	21 ± 1.3	20 ± 0.7	20 ± 0.7
Height (cm)	179 ± 1.8	177 ± 1.5	180 ± 2.6
Weight (kg)	75.8 ± 2.2	77.0 ± 2.5	73.1 ± 2.4
BMI (kg/m^2^)	23.7 ± 1.6	24.6 ± 1.4	22.6 ± 1.9
Fat mass (%) *	14.6 ± 1.0	14.2 ± 0.7	13.3 ± 0.9
Muscle mass (%) **	42.2 ± 2.1	41.2 ± 3.4	42.5 ± 1.9
Heart rate (bpm) ***	166 ± 19.7	156 ± 21.9	155 ± 18.6

According to Siri * and to Lee ** equations, respectively (International Society for Advancement of Kinanthropometry). *** Mean of 3 determinations in each half-time (total = 6 determinations/match), only in field players. Abbreviations used: BMI, body mass index; bpm, beats per minute.

**Table 2 nutrients-15-00177-t002:** Lactate and serum parameters related to muscle damage determined under baseline conditions, pre-match and immediately post-match in the control (CG), almond beverage (AB) and almond beverage + Lippia extract (AB + LE) groups.

Parameter (Units)	CG	AB	AB + LE
Baseline conditions (Day 1)			
Lactate (mg/dL)	8.5 ± 1.1	8.9 ± 0.9	8.2 ± 0.8
Myoglobin (ng/mL)	29.8 ± 2.6	32.2 ± 3.1	33.0 ± 3.4
AST/GOT (U/L)	21.7 ± 2.1	23.3 ± 1.7	22.8 ± 1.9
ALT/GPT (U/L)	21.6 ± 1.9	20.9 ± 2.3	19.9 ± 1.6
CK (U/L)	184.2 ± 17.6	189.4 ± 20.7	191.1 ± 15.5
Pre-match (Day 22)			
Lactate (mg/dL)	8.6 ± 1.5	8.2 ± 2.7	8.0 ± 1.9
Myoglobin (ng/mL)	33.2 ± 2.9	35.3 ± 3.9	34.7 ± 1.7
AST/GOT (U/L)	28.8 ± 1.4 *	25.3 ± 2.9	24.3 ± 2.6
ALT/GPT (U/L)	24.7 ± 2.6	22.7 ± 1.7	22.4 ± 1.6 *
CK (U/L)	214.7 ± 25.1 *	217.5 ± 19.6 *	213.7 ± 22.1 *
Post-match (Day 22)			
Lactate (mg/dL)	13.2 ± 3.8 *^,&^	14.9 ± 2.8 *^,&^	13.7 ± 3.6 *^,&^
Myoglobin (ng/mL)	40.0 ± 3.3 *^,&^	39.8 ± 2.5 *^,&^	38.9 ± 7.2 *^,&^
AST/GOT (U/L)	30.6 ± 3.7 *	25.5 ± 3.8	26.1 ± 4.2
ALT/GPT (U/L)	26.3 ± 3.9	23.8 ± 2.5	25.1 ± 2.9 *
CK (U/L)	278.6 ± 31.8 *^,&^	290.3 ± 34.6 *^,&^	288.1 ± 37.2 *^,&^

* Significant differences (*p* < 0.05) before and after the match compared to baseline conditions (Day 1) in the same group. ^&^ Significant differences (*p* < 0.05) after the match compared to pre-match conditions in the same group. Abbreviations used: AST/GOT aspartate aminotransferase/serum glutamic oxaloacetic transaminase, ALT/GPT alanine aminotransferase/serum glutamic pyruvic transaminase, CK creatine kinase.

**Table 3 nutrients-15-00177-t003:** Presence of oxidative markers in neutrophils, PBMCs and plasma determined under baseline conditions, pre-match and immediately post-match in the control (CG), almond beverage (AB) and almond beverage + Lippia extract (AB + LE) groups.

Marker (Units)	CG	AB	AB + LE
Neutrophils			
MDA (mmmols/L)			
Baseline (Day 1)	21.6 ± 3.3	18.4 ± 2.1	19.6 ± 3.1
Pre-match (Day 22)	19.6 ± 2.8	13.9 ± 2.3 *^,&^	18.7 ± 2.5
Post-match (Day 22)	30.7 ± 3.4 *	22.7 ± 2.3 ^&^	24.1 ± 3.7 ^&^
Protein carbonyls (mmols/L)			
Baseline (Day 1)	12.8 ± 1.9	13.8 ± 1.3	14.5 ± 1.3
Pre-match (Day 22)	14.2 ± 3.0	13.4 ± 2.6	12.6 ± 1.4
Post-match (Day 22)	13.7 ± 2.5	14.6 ± 2.3	15.1 ± 3.1
PBMCs			
MDA (mmols/L)			
Baseline (Day 1)	24.5 ± 4.1	26.1 ± 2.4	23.4 ± 3.3
Pre-match (Day 22)	26.0 ± 3.6	24.9 ± 3.4	20.6 ± 5.8
Post-match (Day 22)	29.9 ± 2.2 *	29.8 ± 3.6	27.7 ± 4.1
Protein carbonyls (mmols/L)			
Baseline (Day 1)	16.7 ± 2.3	14.5 ± 1.4	15.7 ± 1.8
Pre-match (Day 22)	16.8 ± 3.8	15.4 ± 2.2	14.9 ± 1.6
Post-match (Day 22)	18.8 ± 3.4	17.8 ± 2.8	16.3 ± 3.1
Plasma			
MDA (µmols/L)			
Baseline (Day 1)	124.3 ± 15.1	131.8 ± 11.8	128.8 ± 12.3
Pre-match (Day 22)	151.8 ± 19.4 *	110.8 ± 14.1 *^,&^	126.4 ± 13.1 ^&^
Post-match (Day 22)	177.2 ± 18.3 *	141.1 ± 18.7^&^	145.6 ± 21.4 ^&^
Protein carbonyls (µmols/L)			
Baseline (Day 1)	92.6 ± 10.3	88.9 ± 11.1	100.5 ± 9.3
Pre-match (Day 22)	120.7 ± 17.6 *	72.8 ± 7.7 *^,&^	91.1 ± 11.8 ^&,†^
Post-match (Day 22)	137.6 ± 19.6 *	96.3 ± 15.1 ^&^	103.2 ± 20.0 ^&^

* Significant differences (*p* < 0.05) pre- and post-match compared to baseline conditions in the same group. ^&^ Significant differences (*p* < 0.05) compared to CG for the same parameter in the same period of time. ^†^ Significant differences (*p* < 0.05) comparing AB vs. AB + LE for the same parameter in the same period of time. Intracellular malondialdehyde (MDA) and protein carbonyl concentrations were calculated assuming a value of 300 µL/10^9^ cells [16].

**Table 4 nutrients-15-00177-t004:** Activities of antioxidant enzymes in neutrophils and PBMCs and myeloperoxidase in neutrophils determined under baseline conditions, pre-match and immediately post-match in the control (CG), almond beverage (AB) and almond beverage + Lippia extract (AB + LE) groups.

Enzymatic Activity (Units)	CG	AB	AB + LE
Neutrophils			
SOD (pkat/10^9^ cells)			
Baseline (Day 1)	24.1 ± 4.6	31.2 ± 1.6	28.6 ± 3.3
Pre-match (Day 22)	37.9 ± 9.8	26.6 ± 5.1 ^&^	17.8 ± 7.4 *^,&^
Post-match (Day 22)	47.1 ± 7.3 *	38.2 ± 4.5 ^&^	12.3 ± 3.6 *^,&,†^
CAT (k/10^9^ cells)			
Baseline (Day 1)	28.4 ± 2.7	20.8 ± 1.4	27.9 ± 5.6
Pre-match (Day 22)	35.7 ± 8.3	60.2 ± 9.6 *^,&^	62.6 ± 3.6 *^,&^
Post-match (Day 22)	57.3 ± 5.5 *	67.2 ± 8.5 *	62.2 ± 6.2
GPX (nkat/10^9^ cells)			
Baseline (Day 1)	55.6 ± 6.4	53.3 ± 7.8	63.0 ± 9.8
Pre-match (Day 22)	60.3 ± 7.2	52.4 ± 10.2	70.7 ± 9.9 ^†^
Post-match (Day 22)	77.2 ± 9.5 *	79.0 ± 7.5 *	119.1 ± 13.9 *^,&^
GRD (nkat/10^9^ cells)			
Baseline (Day 1)	456.4 ± 23.8	419.7 ± 18.9	420.1 ± 32.7
Pre-match (Day 22)	417.4 ± 28.9	449.3 ± 33.6	434.9 ± 34.1
Post-match (Day 22)	497.1 ± 39.1 *	461.1 ± 39.4	424.0 ± 30.2
MPO (µkat/10^9^ cells)			
Baseline (Day 1)	55.6 ± 4.4	43.6 ± 4.0	52.8 ± 10.9
Pre-match (Day 22)	50.9 ± 5.3	30.9 ± 2.2 *^,&^	39.8 ± 2.0 *^,&^
Post-match (Day 22)	68.7 ± 9.6 *	44.7 ± 3.5 ^&^	47.7 ± 4.2 ^&^
PBMCs			
SOD (pkat/10^9^ cells)			
Baseline (Day 1)	77.1 ± 9.5	73.7 ± 13.1	76.3 ± 6.9
Pre-match (Day 22)	87.2 ± 11.5	89.1 ± 14.3	81.1 ± 9.5
Post-match (Day 22)	77.7 ± 10.4	82.0 ± 7.5	82.8 ± 8.2
CAT (k/10^9^ cells)			
Baseline (Day 1)	95.5 ± 12.5	85.5 ± 12.9	80.6 ± 14.8
Pre-match (Day 22)	107.3 ± 12.5	99.6 ± 23.4	115.2 ± 29.1
Post-match (Day 22)	112.6 ± 19.6	90.6 ± 16.2	98.7 ± 15.6
GPX (nkat/10^9^ cells)			
Baseline (Day 1)	122.6 ± 22.0	125.8 ± 25.6	119.4 ± 24.7
Pre-match (Day 22)	127.8 ± 19.0	121.2 ± 25.1	128.6 ± 23.7
Post-match (Day 22)	137.2 ± 17.3	120.3 ± 20.1	124.2 ± 16.5
GRD (nkat/10^9^ cells)			
Baseline (Day 1)	507.9 ± 44.8	510.4 ± 36.8	519.6 ± 59.5
Pre-match (Day 22)	477.1 ± 49.3	444.4 ± 54.2	462.4 ± 46.8
Post-match (Day 22)	557.1 ± 50.6	642.6 ± 44.4	541.3 ± 52.2

* Significant differences (*p* < 0.05) post-match compared to basal conditions in the same group. ^&^ Significant differences (*p* < 0.05) compared to CG for the same parameter in the same period of time. ^†^ Significant differences (*p* < 0.05) comparing AB vs. AB + LE for the same parameter in the same period of time. Abbreviations used: CAT, catalase; GPX, glutathione peroxidase; GRD, glutathione reductase; MPO, myeloperoxidase; SOD, superoxide dismutase.

## Data Availability

Data were available upon reasonable request to the corresponding authors.

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
