# Peer review of "Modulation of Oxidative Stress and Antioxidant Response by Different Polyphenol Supplements in Five-a-Side Football Players"

_nutrients, 2022, doi:10.3390/nu15010177_

Round 1

Reviewer 1 Report

In this manuscript, Carrera-Quintanar et al. sought to investigate the effects of polyphenolic antioxidant-rich beverages in athletes before/after high intensity exercise (5 vs 5 soccer).
In brief, players were assigned to a control group (water) or 2 intervention groups consuming an almond beverage or an almond beverage enriched with Lippia citriodora extract for 22 days. All groups went through the same training program throughout the study and played a soccer game on the last day.
Overall, players displayed increased markers of inflammation after the game.
However, some of these changes were attenuated in both intervention groups.
The authors posit that supplements could be used for optimal recovery after high intensity exercise.

I truly enjoyed reading the manuscript.
I only have a few minor comments:
- The rationale for using lippia citriodora extract could be made clearer. I am unclear about why it was studied in the first place. Also, it should be defined. What does it contain exactly?
- I appreciate the discussion revolving around the goalkeeping position being somewhat unique. This is a question I had from the beginning and I was thus glad to see that the authors had addressed this point when I got to this section. However, I remain unconvinced about the relevance of including goalkeepers' data with those of "field" players who go through a much more intense aerobic exercise during a game. The nature of their exercise differs greatly. Would the conclusions be different if they were studied independently?
- "Diet adherence was supervised twice a week for each group of participants." How was that monitored? Was that self-reported?
- How did the investigators made sure that the intervention group consumed the totality of their drinks daily?
- The authors attribute the observed effects to the polyphenolic content of the beverages. Could almond proteins and/or fat also play a role in the anti-inflammatory effects?
- The authors assume that these biological parameters translate into post-match recovery. Was this ever measured /established conclusively?

Author Response

Elche, December 23th, 2022

Dear Editor,

We thank you for the opportunity to revise our manuscript entitled “Modulation of Oxidative Stress and Antioxidant Response by Different Polyphenol Supplements in Five-a-Side Football Players” by Lucrecia Carrera-Quintanar, Lorena Funes, María Herranz-López, Néstor Vicente-Salar, Raul Bonet-García, Carles Blasco-Peris, Vicente Micol, Antoni Pons and Enrique Roche. We appreciate the careful review and constructive suggestions made by Reviewer-1. We strongly think that the manuscript has been substantially improved after making the suggested changes.

Following this letter is the rebuttal letter for the reviewer`s comments with our responses. Changes are highlighted in the manuscript in red color for Reviewer-1 and blue color for Reviewer-2.

Sincerely

Dr Antoni Pons and Dr Enrique Roche

COMMENTS OF REVIEWER-1

COMMENT-1: In this manuscript, Carrera-Quintanar et al. sought to investigate the effects of polyphenolic antioxidant-rich beverages in athletes before/after high intensity exercise (5 vs 5 soccer). In brief, players were assigned to a control group (water) or 2 intervention groups consuming an almond beverage or an almond beverage enriched with Lippia citriodora extract for 22 days. All groups went through the same training program throughout the study and played a soccer game on the last day.

Overall, players displayed increased markers of inflammation after the game.
However, some of these changes were attenuated in both intervention groups.
The authors posit that supplements could be used for optimal recovery after high intensity exercise.

I truly enjoyed reading the manuscript. I only have a few minor comments:

ANSWER-1: We really appreciate this general comment. This type of studies is difficult to perform, because the intensity is not the same regarding the role of players in the field (particularly in soccer) as well as the match, more demanding when the opponent is potent. However, studies are necessary in order to advance and give some knowledge that could help players for an optimal recovery. 

COMMENT-2: The rationale for using Lippia citriodora extract could be made clearer. I am unclear about why it was studied in the first place. Also, it should be defined. What does it contain exactly?

ANSWER-2: We have used Lippia citriodora because we have accumulated a solid scientific evidence for the use of this extract as an adequate supplement during the post-exercise recovery process. Publications by our group are listed below:

  • Carrera-Quintanar L, Funes L, Herranz-López M, Martínez-Peinado P, Pascual-García S, Sempere JM, Boix-Castejón M, Córdova A, Pons A, Micol V, Roche E. Antioxidant Supplementation Modulates Neutrophil Inflammatory Response to Exercise-Induced Stress. Antioxidants (Basel). 2020; 9:1242. doi: 10.3390/antiox9121242.
  • Carrera-Quintanar L, Funes L, Sánchez-Martos M, Martinez-Peinado P, Sempere JM, Pons A, Micol V, Roche E. Effect of a 2000-m running test on antioxidant and cytokine response in plasma and circulating cells. J Physiol Biochem. 2017; 73: 523-530. doi: 10.1007/s13105-017-0575-z.
  • Carrera-Quintanar L, Funes L, Vicente-Salar N, Blasco-Lafarga C, Pons A, Micol V, Roche E. Effect of polyphenol supplements on redox status of blood cells: a randomized controlled exercise training trial. Eur J Nutr. 2015; 54: 1081-1093. doi: 10.1007/s00394-014-0785-x.
  • Carrera-Quintanar L, Funes L, Viudes E, Tur J, Micol V, Roche E, Pons A. Antioxidant effect of lemon verbena extracts in lymphocytes of university students performing aerobic training program. Scand J Med Sci Sports. 2012; 22: 454-461. doi: 10.1111/j.1600-0838.2010.01244.x.
  • Carrera-Quintanar L, Funes L, Vicente-Salar N, Blasco-Lafarga C, Pons A, Micol V, Roche E. Effect of polyphenol supplements on redox status of blood cells: a randomized controlled exercise training trial. Eur J Nutr. 2015; 54: 1081-1093. doi: 10.1007/s00394-014-0785-x.
  • Martínez-Rodríguez A, Moya M, Vicente-Salar N, Brouzet T, Carrera-Quintanar L, Cervello E, Micol V, Roche E. Biochemical and psychological changes in university students performing aerobic exercise and consuming Lemon verbena extracts. Current Topics in Nutraceutical Research. 2015; 13: 95-102.
  • Funes L, Carrera-Quintanar L, Cerdán-Calero M, Ferrer MD, Drobnic F, Pons A, Roche E, Micol V. Effect of lemon verbena supplementation on muscular damage markers, proinflammatory cytokines release and neutrophils' oxidative stress in chronic exercise. Eur J Appl Physiol. 2011; 111: 695-705. doi: 10.1007/s00421-010-1684-3.
  • Funes L, Fernández-Arroyo S, Laporta O, Pons A, Roche E, Segura-Carretero A, Fernández-Gutiérrez A, Micol V. Correlation between plasma antioxidant capacity and verbascoside levelsin rats after oral administration of lemon verbena extract. Food Chem. 2009; 117: 589-598. doi:10.1016/j.foodchem.2009.04.059.

The composition of Lippia citriodora extract has been published 12 years ago (Funes et al, 2011). Figure 1 of this publication shows the HPLC profile at 330 nm. The presence of a main peak of verbascoside was a key finding. In a subsequent publication, we demonstrated that verbascoside was an activator of glutathione-dependent enzymes (Carrera-Quintanar et al, 2012). For this reason, Lippia citriodora extract was selected to complete the antioxidant action of the almond beverage. This information has been included in the Discussion (see lanes 384-388).

COMMENT-3: I appreciate the discussion revolving around the goalkeeping position being somewhat unique. This is a question I had from the beginning and I was thus glad to see that the authors had addressed this point when I got to this section. However, I remain unconvinced about the relevance of including goalkeepers' data with those of "field" players who go through a much more intense aerobic exercise during a game. The nature of their exercise differs greatly. Would the conclusions be different if they were studied independently?

ANSWER-3: This a very good question. We were waiting for less oxidative damage in goal keepers compared to field players. However, when we analysed the results we observed no significant differences. As we explain in the Discussion, goal keepers have a more active participation in futsal compared to other soccer variants. In addition, they experience more traumatic and contact actions against the floor and with other players respectively. This was observed in a friendly match (present manuscript) and could be more intense after a competitive official match. At present, this is the only explanation we have to interpret our findings. This is why we express our concern in this respect, in order to aim other scientific groups to collaborate with us or to investigate in this interesting aspect.

COMMENT-4: "Diet adherence was supervised twice a week for each group of participants." How was that monitored? Was that self-reported?

ANSWER-4: At the end of 2 training sessions, the Nutritionist had a meeting with each player. We have designed a pocket book with pictures and following the serving method. At the end of the book, the participant has to note the consumed servings for cereals, legumes, fruits, vegetables, dairy products, protein foods and fats. The book is in Spanish with ISBN: 978-84-947709-3-7 and edited by LIMENCOP SL (Spain). At the same time, the Nutritionist asked for doubts or management of diet to each player. We have completed this information in lanes 129-130.

COMMENT-5: How did the investigators made sure that the intervention group consumed the totality of their drinks daily?

ANSWER-5: During the meeting with the Nutritionist, participants were consuming the beverages. Coach controlled beverage consumption as well the days with no control by the Nutritionist. The resting days, beverages were consumed at home by each player. Of course, we cannot control at 100% the consumption at home, but at least we assured beverage consumption during the training days. We have added this information at lanes 142-144.

COMMENT-6: The authors attribute the observed effects to the polyphenolic content of the beverages. Could almond proteins and/or fat also play a role in the anti-inflammatory effects?

ANSWER-6: The accumulated evidence suggest that polyphenols are the main players in the antioxidant defense. Nevertheless, the comment of the Reviewer-1 is correct and we have considered this possibility for a future research at the end of Discussion (lanes 401-403).

COMMENT-7: The authors assume that these biological parameters translate into post-match recovery. Was this ever measured /established conclusively?

ANSWER-7: This is our main hypothesis that we plan to verify in a new project, studying players of the first division of the Spanish soccer League. Actually, we are waiting for funding, but we have a contact with 2 teams of “La Liga”: Celta de Vigo and Elche.

Reviewer 2 Report

The paper describes the effect of different polyphenolic antioxidant-rich beverages in 2 supplemented groups (almond beverage, AB group, enriched with vitamins C and E and a group supplemented by the same beverage but enriched with Lippia citriodora extract, AB+LE group), of five-a-side/futsal players respect to a control group (CG). Oxidative stress markers and antioxidant enzyme activities in plasma, neutrophils and peripheral blood mononuclear cells (PBMCs) were determined in the three groups at resting conditions and tobefore and after exercise. Supplemented groups present lower oxidative stress markers in neutrophils after exercise but lower antioxidant activities (SOD) and also lower increase in myeloperoxidase activity. The authors conclude that the antioxidant and anti-inflammatory supplements could be usefull for optimal recovery after high intensity futsal games.

Lines 26-27 “Blood extractions were performed at the beginning of intervention,” what is meant by at the beginning of intervention? It means at Day 1?

In Materials and methods the experimental plane was well designed, and explained, in order to homogenize experimental conditions

Lines 138-140 antioxidant potential of the beverages, in addition to the components mentioned, is also due to the orange juice

Results

Table 2 shows the significant differences, for serum parameters, pre and post-match compared to baseline condition (*) (at day 1) and, in the same group, significant differences after the match compared to pre-match (&).

It is not considered the significance of different parameters/markers at baseline, before the match and after the match between CG, AB and AB+LE? Or If not the absolute value, its increment compared to the previous value, between the three groups?

Lines 208-210. Significant increases in CK in all groups and myoglobin in the CG, AB and AB+LE groups confirmed.. What does it mean? it is not clear which groups are being referred to.

Table 3 should be graphically rewritten to better distinguish baseline, pre-match and post match groups.

Line 249 “that which was” it is not correct

Also in Table 4 the 3 conditions baseline, pre and post-match are not well identified

The goal keepers is included in the study but their role is hardly comparable to that of the other players but also, given the small number, difficult to evaluate as a separate role.

Lines 352-353 the muscle damage, represented by CK and myoglobin circulating levels does’nt differ between groups, this result is not discussed

As reported in ref 16  the effect of intracellular antioxidants, eg superoxide dismutase levels were detected when certain supplements were consumed; while high doses of antioxidant supplements taken over long periods of time can impair the antioxidant adaptive response, the antioxidant effect of supplements depends on the dose and the time of intake (line 69). ROS are instrumental agents in the transduction pathway that culminates in the activation of genes coding for antioxidant defenses and metabolic enzymes, where Nrf2-Keap1 signaling plays a key role.

On these basis the results of this study cannot be generalized, for example Lippia extract containing AB+LE group supplement not always reinforce antioxidant effects of AB group. And also supplemented groups not always present a reduction in stress markers, es in PBMC or in neutrophils. Further studies will probably be necessary to clarify the real impact of such supplements in this type of physical activity.

Author Response

Elche, December 23th, 2022

Dear Editor,

We thank you for the opportunity to revise our manuscript entitled “Modulation of Oxidative Stress and Antioxidant Response by Different Polyphenol Supplements in Five-a-Side Football Players” by Lucrecia Carrera-Quintanar, Lorena Funes, María Herranz-López, Néstor Vicente-Salar, Raul Bonet-García, Carles Blasco-Peris, Vicente Micol, Antoni Pons and Enrique Roche. We appreciate the careful review and constructive suggestions made by Reviewer-2. We strongly think that the manuscript has been substantially improved after making the suggested changes.

Following this letter is the rebuttal letter for the reviewer`s comments with our responses. Changes are highlighted in the manuscript in red color for Reviewer-1 and blue color for Reviewer-2.

Sincerely

Dr Antoni Pons and Dr Enrique Roche

COMMENTS OF REVIEWER-2

The paper describes the effect of different polyphenolic antioxidant-rich beverages in 2 supplemented groups (almond beverage, AB group, enriched with vitamins C and E and a group supplemented by the same beverage but enriched with Lippia citriodora extract, AB+LE group), of five-a-side/futsal players respect to a control group (CG). Oxidative stress markers and antioxidant enzyme activities in plasma, neutrophils and peripheral blood mononuclear cells (PBMCs) were determined in the three groups at resting conditions and before and after exercise. Supplemented groups present lower oxidative stress markers in neutrophils after exercise but lower antioxidant activities (SOD) and also lower increase in myeloperoxidase activity. The authors conclude that the antioxidant and anti-inflammatory supplements could be usefull for optimal recovery after high intensity futsal games.

COMMENT-1: Lines 26-27 “Blood extractions were performed at the beginning of intervention,” what is meant by at the beginning of intervention? It means at Day 1?

ANSWER-1: Yes, as indicated in Materials and Methods. We indicated this in the Abstract as well as suggested (lane 27).

COMMENT-2: In Materials and methods the experimental plane was well designed, and explained, in order to homogenize experimental conditions.

Lines 138-140 antioxidant potential of the beverages, in addition to the components mentioned, is also due to the orange juice.

ANSWER-2: The Reviewer-2 is right and we cannot discard this possibility. We have competed this information accordingly (lanes 141-142).

Results

COMMENT-3: Table 2 shows the significant differences, for serum parameters, pre and post-match compared to baseline condition (*) (at day 1) and, in the same group, significant differences after the match compared to pre-match (&).

It is not considered the significance of different parameters/markers at baseline, before the match and after the match between CG, AB and AB+LE? Or If not the absolute value, its increment compared to the previous value, between the three groups?

ANSWER-3: In this particular Table differences between groups were not significant. However, this comparison was made in Tables 3 and 4 where differences in certain parameters were significant when supplemented groups were compared.

COMMENT-4: Lines 208-210. Significant increases in CK in all groups and myoglobin in the CG, AB and AB+LE groups confirmed. What does it mean? it is not clear which groups are being referred to.

ANSWER-4: The Reviewer-2 is right. The sentence is not written appropriately. We refer all the time to all groups (CG, AB and AB+LE). See new sentence in lane 213.

COMMENT-5: Table 3 should be graphically rewritten to better distinguish baseline, pre-match and post-match groups.

ANSWER-5: We have rewritten Table 3 to better compare baseline, pre- and post-match situations. See new Table 3.

COMMENT-6: Line 249 “that which was” it is not correct

ANSWER-6: Corrected (lane 254)

COMMENT-7: Also in Table 4 the 3 conditions baseline, pre and post-match are not well identified.

ANSWER-7: We have rewritten Table 4 to better compare baseline, pre- and post-match situations. See new Table 4.

COMMENT-8: The goal keepers is included in the study but their role is hardly comparable to that of the other players but also, given the small number, difficult to evaluate as a separate role.

ANSWER-8: This a very good question. We were waiting for less oxidative damage in goal keepers compared to field players. However, when we analysed the results we observed no significant differences. As we explain in the Discussion, goal keepers have a more active participation in futsal compared to other soccer variants. In addition, they experience more traumatic and contact actions against the floor and with other players respectively. This was observed in a friendly match (present manuscript) and could be more intense after a competitive official match. At present, this is the only explanation we have to interpret our findings. This is why we express our concern in this respect, in order to aim other scientific groups to collaborate with us or to investigate in this interesting aspect.

COMMENT-9: Lines 352-353 the muscle damage, represented by CK and myoglobin circulating levels does’nt differ between groups, this result is not discussed.

ANSWER-9: We interpret that similar muscle damage occurs in all groups as a result of a similar intensity during the match. The intervention was performed playing a friendly match and intensity was likely similar in all groups. We have explained this aspect in the manuscript (lanes 359-360).

COMMENT-10: As reported in ref 16 the effect of intracellular antioxidants, eg superoxide dismutase levels were detected when certain supplements were consumed; while high doses of antioxidant supplements taken over long periods of time can impair the antioxidant adaptive response, the antioxidant effect of supplements depends on the dose and the time of intake (line 69). ROS are instrumental agents in the transduction pathway that culminates in the activation of genes coding for antioxidant defenses and metabolic enzymes, where Nrf2-Keap1 signaling plays a key role.

On these basis the results of this study cannot be generalized, for example Lippia extract containing AB+LE group supplement not always reinforce antioxidant effects of AB group. And also supplemented groups not always present a reduction in stress markers, es in PBMC or in neutrophils. Further studies will probably be necessary to clarify the real impact of such supplements in this type of physical activity.

ANSWER-10: This is a key point that we have considered in the final Conclusions (lanes 417-418).
